# *Corchorus tridens* L.: A Review of Its Botany, Phytochemistry, Nutritional Content and Pharmacological Properties

**DOI:** 10.3390/plants13081096

**Published:** 2024-04-13

**Authors:** Refilwe Given Kudumela, Thanyani Emelton Ramadwa, Neo Mokgadi Mametja, Tracy Madimabi Masebe

**Affiliations:** Department of Life and Consumer Science, College of Agriculture and Environmental Sciences, Florida Science Campus, University of South Africa, 28 Pioneer Ave, Florida Park, Johannesburg 1709, South Africa; 11837144@mylife.unisa.ac.za (R.G.K.); ramaate@unisa.ac.za (T.E.R.); mametnm@unisa.ac.za (N.M.M.)

**Keywords:** *Corchorus tridens* L., wild edible plants, nutritional content, biological activities, phytochemistry, bioactive compounds

## Abstract

Phytotherapy is a cost-effective alternative that continues to evolve. This has sparked significant research interest in naturally occurring compounds found in edible plants that possess antibacterial, antioxidant, and anticancer properties. *Corchorus tridens* L. is a wild edible plant widely recognised for its edible leaves, which are used for vegetable and animal feed. The plant is widely distributed across the African continent and is utilised in numerous countries for treating fever, pain, inflammation, and sexually transmitted diseases. Extracts from various parts of this plant exhibit antimicrobial, antioxidant, and pesticidal properties. This plant is a rich source of amino acids, vitamins, essential fatty acids, proteins, and minerals, as well as secondary metabolites such as alkaloids, flavonoids, quinines, steroids, terpenoids, phenols, and tannins. Additional studies are still needed to determine other biological activities, such as anti-inflammatory activity, involvement in the treatment of measles, prevention of anaemia, and pain-relieving properties. The current review aims to provide information on the characteristics, distribution, nutritional content, bioactive compounds, traditional uses, and biological activities of the edible plant species *C. tridens* L. to stimulate further research interest to address the existing literature gaps concerning this plant.

## 1. Introduction

Phytotherapy is a highly intriguing alternative that is cost-effective and continues to evolve. Natural chemicals of plant origin, known as secondary metabolites, have attracted significant attention in research for the development of novel therapeutic agents with fewer adverse effects [1]. Moreover, edible plants may be used for their anti-bacterial and anti-cancer properties in the treatment of various ailments and diseases by rural inhabitants [2,3,4]. The consumption of traditional leafy vegetables is crucial because they are rich in a variety of phytochemicals that improve health. Particularly, the southern African region has enormous potential to address obesity and related illnesses while ensuring food security by introducing functional foods based on vegetables [5].

Despite the common use of wild edible plants as sources of food and medicine throughout Africa, there is limited information available about their bioactive compounds, nutritional value, and therapeutic activities. *Corchorus tridens* L., also known as *Thelele* in Sepedi (South Africa), is a wild edible plant species [6,7] widely distributed in 29 African countries, [8] including Zimbabwe [9], Senegal [10] and South Africa [11], where the leaves are consumed as vegetables. This leafy vegetable has several uses in ethnomedicine, including being used as feed for livestock [12,13], and for treating sexually transmitted infections [14], diarrhoea and stomach complaints [15], back pain [16], fever [17], and inflammation [8]. Although limited, the documented in vitro pharmacological evaluation studies of this plant have revealed several biological activities, including anti-oxidant [18], anti-microbial [19,20], and pesticidal properties [21]. Furthermore, the roots, leaves, stems, and fruits of *C. tridens* L. are rich sources of proteins, carbohydrates, crude fibre, vitamins, essential amino acids, essential fatty acids, and essential elements and/or minerals [22,23,24].

This plant is a member of the genus *Corchorus*, which belongs to the Malvaceae family, previously known as Tilliaceae. The genus comprises annual and perennial plants that grow as erect or semi-erect herbs with tender leaves. The genus is reported to contain approximately 100 species, which are widely distributed in tropical and subtropical regions, with some being the source of fibre for jute production [25]. Most of the species are consumed as vegetables; however, roughly half of the species are found in Africa [15,16,17], mostly in the eastern to southern parts of the continent [26], with most species occurring in South Africa [8]. Cultivated species include *Corchorus olitorius* L. [27], *Corchorus capsularis* [24], and *Corchorus incifolious* [28], with *C. olitorius* and *C. capsularis* being widely used for commercial purposes [25]. Wild species include *Corchorus tridens* L. [2], *Corchorus pseudo-capsularis* [29], *Corchorus pseudo-olitorius* [29], *Corchorus aestuans* [29], *Corchorus trilocularis* [2], and *Corchorus fasicularis* [29,30].

The principal taxa in the genus *Corchorus* found in Egypt are *C. olitorius*, *C. capsularis*, *C. trilocularis*, and *C. depressus* [25], with *C. depressus* L., *C. tridens* L., and *C. trilocularis* L. recorded as wild species in that country [2]. Some members of this genus have been reported to contain important bioactive compounds, such as cardiac glycosides, triterpenes, ionones, phenolics, sterols, coumarins, steroids, and fatty acids. These are important constituents of plants used in traditional medicine [31]. Similarly, phytochemical studies of extracts of *C. tridens* L. have revealed the presence of important phytochemicals such as flavonoids, phenols, alkaloids, triterpenoids, and tannins [15,20], which are primarily produced by plants for survival [21]. Additionally, about 21 active compounds have been identified from extracts of this plant, whose known therapeutic activity has been indicated in the literature [20]. However, these claims have not been substantiated by actual in vivo or in vitro studies. *Corchorus* species have been reported to improve male sexual ability and have various biological activities, including cardiovascular, anti-convulsant, anti-oesterogenic, anti-histaminic, anti-cancer, analgesic, anti-pyretic, anti-malarial, and inhibitory effects on nitric oxide production [32].

*C. tridens* L. and two other wild relatives were reported to be resilient to abiotic stressors, including waterlogging and drought [32]. This weed, which occasionally infests sorghum, millet, maize, and tobacco shamba fields, is commonly found in open grasslands, savannahs, and riverine forest areas [6]. In South Africa, there is no record of the plant being cultivated. This is supported by the exclusion of the plant from a list of cultivated useful plants. However, *C. tridens* L. has been included in an inventory record of indigenous and naturalised species used by the Vhavenda people [33]. Benor [34] has mentioned that *C. olitorius* and *C. capsularis* are the only cultivated species in Egypt, while Nair [35] has listed *C. tridens* L. as one of the 285 species being cultivated in the 240 home gardens in Benin [35].

Kumari et al. [17] reviewed the health-promoting properties of the leaves of the *Corchorus* genus, and this review emphasised the need to have more reports documenting the therapeutic activities of species from this genus with respect to its traditional uses since only a few have been studied and documented. Therefore, the aim of this review is to document the botany, ethnomedicinal applications, phytochemistry, nutritional content, and pharmacological properties of the edible plant species *Corchorus tridens* L. Our review presented here will also identify research gaps based on the body of scientific literature available at the time of writing, highlight areas for future research opportunities, and provide data for further comprehensive investigation in the future.

## 2. Distribution and Characteristics

### 2.1. Botanical Description

*C. tridens* L. (Figure 1) is an annual, sub-erect herb that can grow up to 150 cm in height. It has linear–lanceolate long leaves that are up to 9 cm or longer [2], and thin, small stems covered with long, soft hairs. The seeds are angular and oblong, measuring up to 1.8 mm in size, with sharp edges. They have a smooth, black surface, with an obliquely truncate apex and base, and a ruminate coat pattern [2]. The different plant parts (leaves, flowers, stems, seeds, and pods) of this plant are depicted in Figure 1 [36,37]. Its synonyms include *Corchorus burmanni* DC, *Corchorus patens* Lehm, *Corchorus senegalensis* Juss. Ex Steud, *Corchorus tridens* var. *euryphyllus Domin*, and *Corchorus trilocularis* Burm.f. [2,36].

The plant is usually found in habitats such as savannah, which is often burned, open plains, open grassland, overgrazed bushland, dry bushland with lava hills, scrub on hillsides, valley floors, riverine forest floors, woodland edges, riparian medium forests, riverbanks, swamps, and waters’ edge [8]. The plant typically grows in various types of soils, including sandy granite, sandy and muddy alluvium, anthill, poorly drained whitish silt, poor heavy red soil, moist mud, bare brown sand, sand between lava outcrops, shingle, pebbles in empty riverbeds, and more. It is described as an uncommon weed in crops such as maize, millet, sorghum, tobacco shambas, and kenaf. It typically occurs in cleared or disturbed soil associated with cultivation [8]. *C. tridens* L. is a naturalised exotic plant that is often gathered as wild spinach in the stress-exposed areas of South Africa characterised by elevated oxygen levels and increased nitrogen mineralisation rates [7,38]. The plant is available all year; however, it is predominately present in the wet season and scarce during the dry seasons [38]. It germinates spontaneously, grows rapidly, has prolific seed production in temperate conditions, and is commonly found in rural communities [38].

### 2.2. Taxonomy

*C. tridens* L., (Figure 1), also known as Horn-Fruited jute in English, Delele in Tshivenda, *Thelele* in Sepedi (South Africa), *Derere* in Shona (Zimbabwe), *Tege* in Cameroon, *Otege* in Luganda (Uganda), Naruvalli in Tamil (India), and *Okalyaoipute* in Oshiwambo (Namibia), is a wild edible plant species. It belongs to the Plantae kingdom, Tracheophyte phylum, Magnoliopsida class, Malvales order, a monophyletic group within the Malvaceae family previously known as Tilliaceae, and the *Corchorus* L. genus. This genus comprises more than fifty (50) species that abundantly grow in the wild [7,17,27,37].

### 2.3. Variation and Distribution

Most species in this genus are morphologically similar, and identifying them during the vegetative phase has proven to be challenging [31]. Although these species may mostly be identified during the reproductive stage using seed and capsule characteristics, the following morphological characters can be used to aid identification in the vegetative stage: shade, the presence or absence of stipules, and the presence or absence of setae, a structure resembling a tail at the base of the blade [31]. *Corchorus tridens* L. and *Corchorus aestuans* are closely related, with a similarity index of almost 100%, and both have small seed morphological characteristics [25].

*C. tridens* L. is widely distributed in most African countries, including South Africa, Senegal, Namibia, Benin, Angola, Guinea-Bissau, Botswana, Burundi, Cameroon, Chad, Egypt, Ethiopia, Kenya, Ghana, Malawi, Somalia, Mozambique, Niger, Mali, Uganda, Togo, Zambia, Sudan, Zimbabwe, Burkina Faso, Tanzania, Nigeria, Mauritania, and Zaire [3]. Furthermore, there are reports mentioning the presence of this plant in India and Australia [17].

## 3. Traditional Uses

Traditional medicine is utilised by many countries in Africa, Asia, and Latin America to meet some of their primary healthcare needs. More than 20,000 plant species are employed for medicinal purposes by different human civilisations worldwide [9]. This is despite the expansion and growth of the pharmaceutical industry [39]. Furthermore, the identification and documentation of wild food plants with medicinal value could aid in the development of new medications derived from these plants, specifically the leaves which can be sustainably collected [40]. Wild edible plants (WEPs) have been shown to be effective in treating a variety of illnesses, including diabetes, nausea, constipation, inflammation, sexually transmitted infections, fever, and malaria [41]. *C. tridens* L. leaves have been used as a vegetable and soup herb for many years, and this is well documented in many countries. Fresh or dried leaves are used as vegetables for humans, while whole plants are used as feed for cattle, goats, and sheep [22,23]. The fibre derived from this plant is strong, long, shiny, and of high quality. It has been utilised for fishing lines in northern Nigeria and other regions [31,34]. The same plant was also reported to have health benefits [42], with ethnomedicinal applications for the treatment of stomach complaints, inflammation, sexually transmitted infections, pain, and fever [17,27,36,37,38]. Generally, the entire plant or various parts are crushed, mixed with water, and then either cooked or consumed as a decoction or infusion for oral treatment. It can be used as a paste for external applications and in the treatment of stomach pains [15], fever [20], and genital ulcers [43]. The plant can also be mixed with *Terminalia sericea* and *Cassia petersiana* for the treatment of sexually transmitted diseases [14]. More detailed information about the traditional uses of *C. tridens* L., together with the mode of preparation and administration, is outlined in Table 1. 

## 4. Chemical Constituents of *Corchorus tridens* L.

### 4.1. Phytochemical Content

Phytochemical compounds are secondary metabolites produced by plants. These compounds are not essential for plant growth and development but are solely responsible for defence against biotic and abiotic stress [44]. These metabolites are found in several plant parts, including roots, leaves, bark, flowers, and seeds, and are distinguished by their great structural diversity [41]. Different classes of compounds categorised as phytochemicals include but are not limited to phenolics, flavonoids, cardiac glycosides, sterols, coumarins, tannins, terpenoids, and alkaloids. These are the active ingredients that have anti-oxidant, anti-cancer, anti-microbial, and immunity-stimulating properties [18].

A study conducted by Shatri and Mumbengegwi [15] aimed at documenting the ethnomedicinal uses and quantifying the contents of phenols and flavonoids in selected plants used to treat gastrointestinal conditions and other ailments in the Likokola village, Omusati region, in Namibia. The study quantified phytochemical compounds such as the total flavonoid and phenolic content in the water and methanolic whole plant extracts of *C. tridens* L. Moreover, the water extracts showed a higher phenolic content than the methanol extracts did. These compounds are well known for having anti-viral, anti-bacterial, and anti-inflammatory effects [44]. Isaiah et al. [20] in a study that was aimed at phytochemical screening, the determination of anti-microbial activity, and a gas chromatography-mass spectrometry (GC-MS) analysis of *Corchorus tridens* L., tested for the presence of alkaloids, amino acids, carbohydrates, flavonoids, glycosides, phenol, proteins, quinones, resins, saponins, starch, steroids, tannins, terpenoids, and vitamin C in ethanol and petroleum ether whole plant extracts of *C. tridens* L. The presence of alkaloids, flavonoids, quinones, steroids, and tannins was seen in both the petroleum ether and ethanol extracts, while amino acids, starch, glycosides, saponins, proteins, or resins were observed in the same extracts. Carbohydrates, vitamin C, and terpenoids were only present in petroleum ether extracts, while phenols were only observed in the ethanolic extract. The presence of functional groups of these active compounds was also confirmed via Fourier-transform infrared spectroscopy (FT-IR) analysis of the extracts [19]. Table 2 summarises the phytochemicals found in different plant part and extracts of *C. tridens* L., together with the analysis method employed. Although this study reported the absence of amino acids and proteins in the whole plant extracts of *C. tridens* L., Freiberger et al. [23] reported the presence of these nutrients in the leaves of *C. tridens* L. These variations could be attributed to various factors, such as the differences in the detection methods used, environmental conditions in the various study areas, the age of the plant samples used, and the timing of harvest. Similar reasons were also provided by Datta et al. [42].

### 4.2. Bioactive Compounds Identified

To the best of our knowledge no bioactive metabolites have been isolated yet from the *Corchorus tridens* L. herb; however, about twenty-one compounds have been identified from the petroleum ether whole plant extract of this plant using GC-MS analysis, and these are listed in Table 3 together with their chemical structures and molecular weight [20]. A study conducted using Fourier-transform infrared (FT-IR) spectroscopy on whole-plant extracts of *C. tridens* L., in ethanol and petroleum ether, also verified the presence of functional groups such as amide, alcohol, phenols and halogen compounds [20]. Different compounds from plants are responsible for various biological activities and in most cases, these are linked with ethnomedicinal applications of the plant. n-Hexadecanoic acid, a compound identified from the petroleum ether extract of *C. tridens* L., is known to exhibit anti-oxidant and pesticide activity [20]; this compound could be responsible for the reported anti-oxidant [18] and pesticide activity [21]. It would be interesting to isolate, characterise, and test the compound for the abovementioned activities and perform biological activity assays of the isolated compounds to confirm the preliminary documented results. The same can be carried out for β-amyrin, which exhibits anti-inflammatory activities, and this can be tested and documented since there are no records of anti-inflammatory activity studies on the whole plant or any part of *C. tridens* L. to validate the ethnomedicinal application in the treatment of pain and inflammation.

### 4.3. Nutritional Contents

The African continent boasts distinctive floral diversity [39], and many human societies living on the continent continue to depend on a variety of wild edible plants to fulfil a significant portion of their nutritional requirements [43], even with the presence of staple foods and substantial agricultural investments [46]. This is probably because they contain essential nutritional ingredients, such as carbohydrates, proteins, minerals, vitamins, micronutrients, fibres, and fats, which are crucial for the needs of the human body and are employed in a variety of physiological, metabolic, and morphological processes [12,47,48]. This promises to alleviate the food security and nutrition issue that the globe is currently confronting, particularly in sub-Saharan African countries that heavily depend on imports [12]. Wild edible plants are often consumed as a complement to staple foods and to compensate for food shortages during drought and famine [12]. They include leafy shoots, fruits, seeds, underground organs, and flowers [47]. These plants are easily accessible and can be bought in the market or collected in the wild or in cultivable areas by individuals [23]. The same concept is adapted for the leaves of *C. tridens* L., *Leptadenia hastata*, *Hisbiscus sabdarifa*, and *Moringa oleifera*. When cooked and flavoured with groundnuts, they are often included in midday meals in southern Niger [23].

There is increased interest in evaluating the nutritional properties of wild edible plants in relation to their use. This interest promises to provide more insights into the role of wild edible plants in the nutritional well-being of the community [23]. The nutritional contents, together with the method of analysis used for each plant part, are summarised in Table 4. A study was conducted by Gwarzo et al. [22] to determine the elemental composition of the roots, stems, fruits, and leaves of *C. tridens* L., using neutron activation analysis (NAA) and energy-dispersive X -ray fluorescence analysis (EDXRF). The results revealed no significant difference in the concentrations of elements such as Barium (Ba), Calcium (Ca), Iron (Fe), Potassium (K), Rubidium (Rb), Cobalt (Co), Manganese (Mn), and Vanadium (V) in the various plant parts assayed using both techniques [22]. Furthermore, the study revealed the presence of higher concentrations of potassium ranging from 8985 to 54,290 ppm and of calcium ranging from 3788 to 17,765 ppm. These concentrations are significantly higher than the normal ranges of 10 to 100 ppm for potassium and 50 to 60 ppm for calcium typically found in most plants [22]. These are crucial elements in plants responsible for growth, development, cell wall, and membrane structural roles [49,50]. The same study also reported high amounts of elements such as Ba (8.825–138.0 ppm), Fe (376–8945 ppm), Mn (23–496.224 ppm), Zinc (14.0–51.0 ppm), Rb (3.9–23.0 ppm), and Co (0.00–0.41 ppm). The above-mentioned quantities fall within the typical ranges of Fe (40.0–500.0 ppm), Ba (9.2–131.9 ppm), Mn (50.0–356.0 ppm), Zn (15.0–100.0 ppm), Rb (0.2–194.0 ppm), and Co (0.1–0.60 ppm) that are present in most plants [22]. Moreover, a study by Freiberger et al. [23] analysed the minerals, amino acids, proteins, and fatty acids of the leaves of seven wild edible plant species in southern Niger: *Corchorus tridens* L., *Ximenia americana*, *Maerua crassifolia*, *Moringa olifera*, *Leptadenia hastata*, *Amarunthus viridus*, and *Hibiscus sabdarifa*. Some of the plant species were bought in marketplaces, whereas others were foraged from nearby uncultivated wild plants [23]. The study revealed that all seven wild plants had all the tested minerals, that is, Ca, Fe, Magnesium (Mg), Copper (Cu), Manganese (Mn), Zinc (Zn), Phosphorus (P), and Selenium (Se), in varying quantities. It is worth noting that *C. tridens* L. had the highest quantities of P (4.35 µg/g of dry weight) and Cu (12.8 µg/g of dry weight) and the lowest quantities of Ca (10.2–11.9 µg/g of dry weight) and Se (14.9 µg/g of dry weight) [23]. Regarding protein and amino acid contents, although *C. tridens* L. had the highest quantity of total protein (19–25% of dry weight), it scored 90% and 94% for methionine/cysteine and lysine essential amino acids, respectively, when compared with the World Health Organisation (WHO) standard [23]. Proteins are made up of amino acids, and these are crucial macronutrients that support structural and mechanical function, control cellular and bodily functions, and offer energy when required [51]. In general, essential fatty acids, such as linoleate (18:2n6) and α-linoleate (18:3n3), were present in lower concentrations in all the plant species examined; however, *C. tridens* L. contained the highest amount of linoleate (3.10 mg/g) [23]. The leaves of *C. tridens* L. contains about 4.27 mg/100 g of ash content; although slightly lower than that of other leafy vegetables, this gives an idea of the index of mineral components in a sample [24]. The nutritional contents present in the different parts of *C. tridens* L., together with the amounts and methods used for analysis, are outlined in Table 3. Although important for human metabolism, the body cannot produce these fatty acids; hence, humans must consume foods that contain them [52]. Therefore, *C. tridens* L. is positioned as a source of essential fatty acids with fewer calories and low cholesterol, and possibly contains little to no allergens. The observations from these studies suggest that *C. tridens* L. can serve as an essential source of micronutrients, carbohydrates, crude fibre, vitamins, proteins, essential amino acids, and essential fatty acids for humans and animals. Previous studies have also shown that edible species of the genus *Corchorus* are excellent suppliers of vitamins A, C, and E, as well as minerals such as calcium and iron [28,53,54].

## 5. Biological Activities

Many underdeveloped nations around the world, especially those from rural areas, have access to affordable and readily available medications owing to wild plants [12], which are considered valuable sources of molecules with therapeutic potential [55]. This therapeutic potential is due to the presence of diverse bioactive constituents known as secondary metabolites that possess different mechanisms of action [56]. These secondary metabolites greatly influence the biological effects of therapeutic plants, such as their hypoglycaemic, anti-diabetic, anti-oxidant, anti-microbial, anti-inflammatory, anti-carcinogenic, anti-malarial, anti-cholinergic, and anti-leprosy activities [57]. Different extracts of *C. tridens* L. have been reported to exhibit anti-microbial and anti-oxidant activities [18,19,20]. However, there are relatively few studies on the medicinal and/or biological activities of *C. tridens* L., considering that the keywords “medical properties” and/or “biological activities” in the internet literature search for the current review produced relatively few results. Similar observations were made by Ushie et al. [18] on the scarcity of information related to the biological activities of *C. tridens* L. This is probably because this plant is mostly used as a vegetable and feed for livestock; therefore, its consumers are not aware of any medicinal and/or therapeutic effects it may have. The biological activities mentioned below include anti-microbial, anti-oxidant, and pesticidal properties, which could justify some of the traditional and medicinal applications of this plant, such as the use of the plant for the treatment of stomach ailments and diarrhoea.

### 5.1. Anti-Microbial Activities

Anti-microbial drug resistance continues to threaten the effectiveness of existing synthetic anti-microbial drugs, thus necessitating the need for development of novel molecules with enhanced activity, preferably from plants [17,58]. Plant-based anti-microbial compounds can be used alone or in conjunction with conventional anti-biotics to improve their potency against various resistant microorganisms [59]. Several studies have demonstrated notable anti-microbial activity using different methods for screening plant compounds, such as phenolics, alkaloids, flavonoids, triterpenes, and steroids [59]. Similarly, Kapoor and colleagues [19], in their study titled anti-microbial screening of some herbal plants of the Rajasthan dessert: overview, screened 12 herbal plants, including *C. tridens* L., for anti-microbial activity, and documented the anti-microbial activity of the ethyl ether and alcoholic leaf extracts of *C. tridens* L. against bacterial pathogens (*Staphylococcus aureus* and *Escherichia coli*) as well as the fungal pathogen *Candida albicans* [19]. Although the ethyl ether extract exhibited the highest anti-bacterial activity against *S. aureus* (0.67 I/C^a^), the alcoholic extract of the same plant exhibited the highest activity against *C. albicans* (0.73 I/Ca) [19]. Isaiah et al. [20] also reported the anti-bacterial activity of ethanol and petroleum ether whole-plant extracts of the same plant against *Staphylococcus aureus, Bacillus subtilis, Klebsiella aerogenes*, and *Escherichia coli* using the disc diffusion assay. The highest activity was seen in *E. coli* with the lowest zone of inhibition (12 mm); this was even higher than that of ciprofloxacin, an anti-microbial reference standard whose zone of inhibition was 35, 40, 30, and 38 mm for *Staphylococcus aureus, Bacillus subtilis, Klebsiella aerogenes*, and *Escherichia coli,* respectively. The overall anti-microbial activity seen in the different extracts of *C. tridens* L. could be due to the bioactive compounds, such as phenolics, flavonoids, alkaloids, terpenoids, and tannins, which are known to exhibit anti-microbial activity and have been observed in *C. tridens* L. Some of the solvents listed above as solvents with the ability to extract anti-microbial compounds have been used in two different studies; therefore, this allowed for the successful extraction of these compounds hence, the observed activity. Gram-negative bacteria typically have a more complex cell envelope than their Gram-positive counterparts do; this serves as a diffusional barrier and lessens their susceptibility to antibacterial substances [60]. Therefore, *C. tridens* L. extracts are a promising reservoir for anti-bacterial compounds to combat even the less susceptible Gram-negative bacteria [61]. Both studies employed the disc diffusion method of testing instead of dilution methods such as broth micro-dilution, which allows for the determination of the minimum inhibitory concentration and a differentiation between bacteriostatic and bactericidal effects [62].

### 5.2. Anti-Oxidant Activities

Free radicals, reactive oxygen species (ROS), and reactive nitrogen species (RNS) are produced by cells due to regular metabolic processes, increased exposure to the environment, and the consumption of high concentrations of xenobiotics [63]. In many pathophysiological situations, free radicals are the causative agents of oxidative stress, which changes cellular components, leading to numerous diseases, including atherosclerosis, cancer, diabetes, and liver cirrhosis. Although effective, the safety of synthetic anti-oxidants such as butylated hydroxytoluene (BHT), butylated hydroxyanisole (BHA), propyl gallate, and tertbutylhydroquinone is worrisome since they have been linked with toxicity and carcinogenicity [64]. It is for this reason that anti-oxidants are then sourced externally from natural sources such as medicinal plants and plant-based foods with vitamin A, E (alpha tocopherol), and C (ascorbic acid), minerals, and polyphenols that are known to have anti-oxidant activity [65,66]. Edible plants have biologically active chemical compounds, produced during regular metabolic processes, that have been thoroughly studied for their potential as natural anti-oxidants [67]. Phenolics, flavonoids, and tannins, which are present in many plant components like the leaves, roots, stem bark, seeds, and fruits, are the active substances that can block excessively synthesised free radicals and so serve as anti-oxidants [67]. The polarity of the solvent chosen for the extraction of bioactive compounds from plants is influenced by the type and polarity of the target compounds [64]. As such, polar solvents, such as water, methanol, ethanol, and solvents of intermediate polarity like acetone, are suitable for the extraction of anti-oxidant compounds with reducing properties and free radical scavenging activity [68,69]. Considering this, Ushie and colleagues [18] conducted a study and determined the anti-oxidant activities of n-hexane, ethyl acetate, acetone, and methanol leaf extracts of *C. tridens* L. using the DPPH (1,1-Diphenyl-2-Picrylhydrazyl) assay. The study revealed that all the extracts had radical scavenging activities with the scavenging percentage ranging from 07.54 to 65.83% and the with half maximal inhibitory concentration (IC_50_) ranging from 34.38 to 66.38 mg/mL; this increased with increasing concentrations (0.313–0.500 mg/mL). Methanol is a polar solvent and normally dissolves polar compounds. Extracts extracted with this solvent exhibited the highest inhibitory effect with an IC_50_ concentration of 34.38 mg/mL, which was comparable to that of vitamin C (IC_50_ = 22.52 mg/mL), which was used as a positive control. This is probably because most phytochemicals that act as free radical scavengers are polar molecules that can be extracted using polar solvents; therefore, the types and amounts of phytochemicals that each solvent was capable of extracting based on its polarity may account for the differences in scavenging capacities [18]. Overall, the anti-oxidant activity observed in the extracts of *C. tridens* L. could be linked to phenols, vitamin C, and flavonoids whose presence have been reported in some extracts of this plant [20]. Generally, plants are evaluated using different testing methods, and their mode of anti-oxidant activity is classified as either reducing agents, hydrogen donors, singlet oxygen quenchers, or metal chelators and later divided into primary (chain-breaking) and secondary (preventive) anti-oxidants [70]. Therefore, it would have been interesting to use more than one method for the evaluation of anti-oxidant activity in the different extracts of *C. tridens* L. to establish if there is more than one different mode of anti-oxidant activity, as mentioned above.

### 5.3. Pesticidal Properties

Pests often threaten agricultural crops, and this has an impact on their development and quality; thus, to overcome this, crop farmers rely on the use of synthetic pesticide [71]. Despite their effectiveness, pesticides are commonly associated with pest resistance, persistent residues, nontarget toxicity, and environmental hazards [72]. It is for this reason that plants with bioactive metabolites are positioned as better alternatives for pest management approaches [72] since they are biodegradable, less toxic and have a different mode of action [71]. Gotyal et al. [21] undertook a study to determine the extent of resistance amongst wild and cultivated jute species based on comparative egg laying preferences, biology, and the establishment of the biochemical foundation of resistance against jute hairy caterpillar, i.e., *Splisoma obliqua*. The study reported that the wild species, i.e., *C. tridens* L. and *C. aestuans*, had maximum antibiosis effects on the larval growth and development of *S. obliqua*, which was indicated by the larval weights at day 5 after feeding, which ranged from 3.3 ± 5.8 to 154.0 ± 13.1 mg, the survival percentage, and larval death at day 9. These were significantly lower than those of *C. olitorious* (268 ± 11.2 mg). The former species were also the least preferred for oviposition preferences with the smallest mean number of eggs per cluster (77. 2 ± 8.2 and 75.2 ± 38.8 respectively) compared with that of 174.0 ± 2.7 on *C. olitorious*. These observations were explained by the negative and positive correlation between protein content and phenol content on larval growth, i.e., the wild species that had the highest phenol content and lowest protein content allowed for slow larval growth. The observed activity suggests that *C. tridens* L. has pest resistance and pesticidal properties against *S. obliqua*, which might be because of the presence of secondary metabolites, such as terpenoids, alkaloids, phenols and quinones, which are produced by plants in response to plant injury and which alter herbivore eating, growth, and survival [21]. These substances were reported in the ethyl ether extract of *C. tridens* L. [20]. Therefore, the extracts of *C. tridens* L. should be evaluated further for the development of natural pesticides that are non-toxic and biodegradable, and these can be incorporated in current pest management strategies. The pharmacological activities, together with the type of study conducted, the reference standard used, and the plant extract studied, are summarised in Table 5.

## 6. Toxicology and Clinical Trials

To the best of our knowledge, no documented in vitro or in vivo toxicity research utilising *C. tridens* L. extracts exist. Consequently, no clinical trials have been conducted, because toxicological data serve as a foundation and lead for clinical trials. However, there has been documentation of the in vitro toxicological evaluation of some species in the *Corchorus* genus. These include *C. olitorious*, which was evaluated for cytotoxicity against non-cancerous L929 cell lines against cisplatin as a reference standard. The results showed that out of all the studied extracts, the ethyl acetate extracts had the highest level of toxicity with the lowest percentage of cell viability at the highest concentration of 250 µg/mL (31.70 ± 0.007) [73]. This suggests that extracts of some members of the *Corchorus* genus could be potentially toxic; therefore, caution should be exercised when consuming them.

## 7. Search Procedure

The information used to write this review was gathered from reputable databases (Google Scholar, Science Direct, Scopus, Sprinter Link, Semantic Scholar, ProQuest, PubMed, and PubChem) in accordance with the Preferred Reporting Items for Systematic Reviews and Meta-Analyses (PRISMA) guideline, as described by Moher et al. [74] There were cross-references between *Corchorus tridens* L. and terms such as botany, characteristics, distribution, traditional uses, toxicity, clinical trials, nutritional content, biological activities, anti-oxidant activity, anti-microbial activity, and bioactive compounds. To accommodate the needs or constraints of the database being used, search definitions were executed separately or in limited combinations. In total, 157 entries were returned, and these were screened based on the illegibility criteria below. Out of these, 56 studies in total, published from 1990 to 2023, were selected to document the characteristics, distribution, medicinal importance, traditional uses, biological activities, phytochemical and nutritional contents, and bioactive compounds of the plant species.

### 7.1. Inclusion and Exclusion Criteria

Article titles and abstracts were critically evaluated and screened to remove, where necessary, any duplicated studies with information or research studies that were not related to the aim of this review.

#### 7.1.1. Inclusion Criteria

Access to full-text articles in English.Published studies with information on the botany, distribution, taxonomy, toxicity, pharmacological activities, traditional uses, nutritional content, phytochemical content, and bioactive compounds of *Corchorus tridens* L. and other species within the same genus were included when reviewing the literature for this review paper.

#### 7.1.2. Exclusion Criteria

Articles that were not published in English were excluded.Articles that did not have any information on the plant were excluded.Studies that only listed the plant’s name without providing details on its botany, distribution, traditional uses, phytochemistry, pharmacological and biological effects, nutritional value, or other purposes were excluded.

### 7.2. Risk Bias Assessment

Each included study’s risk of bias was separately evaluated by two reviewers, and conflicts were settled by consulting a third and fourth reviewer. To provide accurate data and foster transparency, these points were used to evaluate the methodological quality of investigations. Excluded studies had levels of bias that were far too high. Studies with significant mistakes in their design, analysis, or reporting, with a significant number of missing data, or with inconsistent reporting constituted a highly unacceptable level of bias.

## 8. Limitations

This study focused on reviewing the available literature on the botany, characteristics, distribution, traditional uses, pharmacological activities, phytochemistry, nutritional content, and biological activities of the plant with potential applications in the health, agriculture, and nutrition industries. Although effective, the methods used in the published studies reviewed are not comprehensive, current, and quantitative, which made the review seem like it only focused on qualitative analysis. This plant has potential; however, there are other important aspects that still need to be addressed to leverage its inherent capabilities. This includes thoroughly investigating the biological activities, such as anti-viral, anti-inflammation, cytotoxicity, anti-microbial activity, against pathogens responsible for sexually transmitted infections with respect to the traditional uses. This could play a role in influencing further research on the plant to explore its potential application in different industries.

## 9. Future Perspectives

*C. tridens* L. is an edible plant widely distributed across the African continent. The plant is commonly consumed as a vegetable paired with a starchy meal. This has prompted evaluations of its nutritional content, which have revealed the presence of different important health-promoting nutrients. These support bodily functions, provide energy, and are responsible for growth and development. This plant has prominent traditional uses particularly in ethnomedicine, and these include treatment for stomach ailments, fever, STI, genital ulcers, measles, and pain, most of which remain to be fully explored for scientific validation through modern scientific research methods. Although in-depth studies are still required, some authors have tried to link some of the ethnomedicinal applications with in vitro therapeutic activity. These include anti-bacterial activity, especially against *E. coli*, which is usually involved in most stomach complaints and diarrhoea cases. The present review has outlined the characteristics, distribution, medicinal significance, traditional applications, biological activities, phytochemical, and nutritional components, as well as bioactive compounds of *C. tridens* L. The documented secondary metabolites include polyphenols, terpenoids, glycosides, saponins, and vitamins that may be responsible for the observed medicinal and pesticidal properties. Furthermore, specific compounds such as Tetracontane, 3,5,24-trimethyl, Tritetracontane, β-amyrin, Stigmastan-6,22-dien, 3,5-dedihydro, and Cholesterol 3-*O*-[(2-acetoxy) ethyl] are linked to anti-inflammatory, anaesthetic, and anti-atherosclerosis effects. Given the increased demand for novel medications, further studies are required to isolate, characterise, and thoroughly study the extracts of this plant species since it holds an untapped reservoir of bioactive compounds. However, before this can happen, there are numerous research gaps that should be filled. Firstly, in vitro and in vivo studies of the extracts of the different parts of the plant should be conducted to link the claimed traditional uses and applications to therapeutic activities, such as the treatment of inflammation, pain-reliving properties, cooling, the prevention of anaemia, and the treatment of genital ulcers and STIs since based on the literature search there is no record of such evaluations yet. Secondly, there is a dearth of information in the literature on the toxicity of this plant both in vitro and in vivo; therefore, this necessitates the need for such evaluations since the plant is used for medicinal purposes, cattle feed, and human consumption. The plant should also be evaluated for the presence of any toxic heavy metals since these have a cumulative tendency and persistent nature in wild vegetables. Thirdly, in clinical trial studies following in vitro and in vivo toxicity evaluation studies, the results obtained could form the basis and guides for clinical trial evaluations of extracts from different parts of this plant. Fourthly, owing to its nutritional content, the plant has potentials for application in the food industry, and this should be explored and evaluated for inclusion in the mainstream food industry. Lastly, active compounds from any extract of this plant are yet to be isolated, characterised, and fully evaluated at a molecular level to determine the mechanism of action of the observed therapeutical activity. Presently, only 21 active compounds have been identified from one whole plant extract; therefore, more extracts should be explored with modern analytical techniques such as ultra performance liquid chromatography mass spectrometry (UPLC-MS), gas chromatography mass spectrometry (GC-MS), and/or ultra-high performance liquid chromatography-mass spectrometry/mass spectrometry (UHPLC-MS/MS).

## 10. Conclusions

In conclusion, *C. tridens* L. is an understudied plant that could have applications in the health and nutrition industry owing to its abundant phytochemicals and nutritional constituents. Therefore, more research is needed to explore the pharmacological properties in relation to its underappreciated traditional usage, particularly in the treatment of STIs and inflammation, prevention of anaemia, and treatment of measles and pain-reliving properties. Our research group is currently working on the plant to pharmacologically evaluate the unexplored therapeutic activities in relation to their traditional uses to fill these identified gaps. Future research should clarify the precise mechanisms of action of *C. tridens* L., and its main bioactive phytochemical constituents. The development of safe and effective dosage forms from this plant requires the completion of clinical trials to assess the pharmacokinetics, safety, appropriate dosage, and efficacy of *C. tridens* L.

## Figures and Tables

**Figure 1 plants-13-01096-f001:**
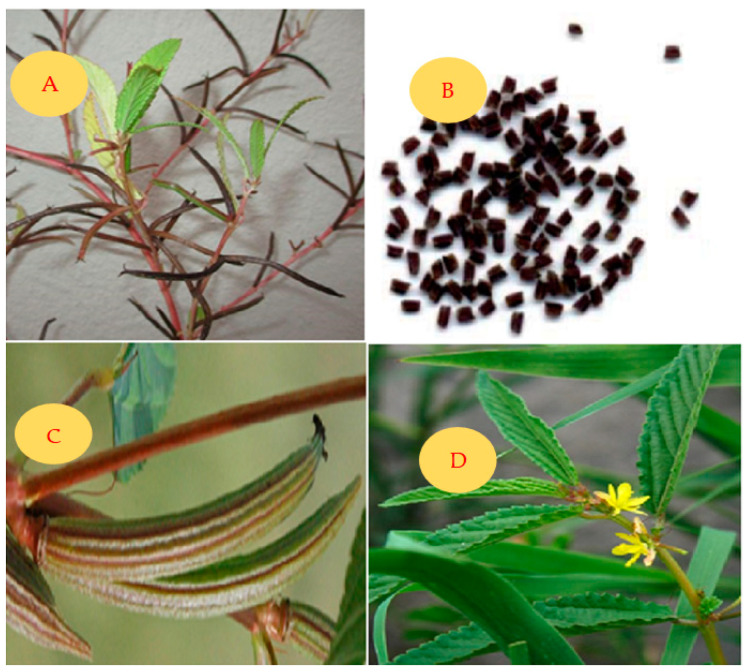
Representation of various parts of *C. tridens* L.: stems (**A**), seeds (**B**), pods (**C**), and leaves and flowers (**D**).

**Table 1 plants-13-01096-t001:** Traditional uses of *Corchorus tridens* L., mode of preparation and administration.

Part Used	Uses	Preparation	Administration	Country/Region	References
Whole plant	Treatment of stomach pains, diarrhoea	Infusion	This is taken orally as a drink	Namibia (Likokola)	[15]
Whole plant	Used for treatment of gonorrhoea and syphilis	Decoction is mixed with *Terminalia sericea* and *Cassia petersiana*	The decoction is taken orally as a drink	South Africa (Venda)	[14]
Whole plant	Used for treatment of measles	Decoction and steaming	The decoction taken orally, and the steam is inhaled	Uganda (Kampala)	[44]
Roots	Used for treatment of back pain	Crushed, mixed with hot water,	The extract is taken orally as a drink	Zimbabwe (Nhema)	[16]
Leaves and young shoot	As vegetable and soup herb	Cooked	This is eaten as a meal paired with a starch	Nigeria, South Africa, Uganda and Zimbabwe	[22]
Leaves	Used to treat burns, cuts, and syphilis sores	Crushed	This is applied directly to the burns, cuts, and sores	South Africa (Limpopo province)	[8]
	Used as a plaster to reduce swellings	Chewed	This is applied directly to the affected area	Tanzania (Musoma)	[38]
Fruits	Used to treat jaundice and sexual problems	Juice	The plant is made up to a juice with water and this is taken orally as a drink	Pakistan (Punjab)	[41]
Roots, leaves, and whole plant	Used for treatment of fever, genital ulcers and to prevent anaemia.	Decoction	This is taken orally as a drink	Zambia (Livingstone), India	[17,20]
Stems and leaves	Used to treat genital ulcers caused by syphilis or chancroids.	Paste	Crushed in water and applied on wounds	Namibia (Ohangwena)	[17,20,45]

**Table 2 plants-13-01096-t002:** Qualitative phytochemical contents of extracts from *C. tridens* L.

Plant Part	Solvent Used	Phytochemicals	Analytical Methods Used	References
Whole plant	Water and methanol	Total phenol and flavonoid contents, coumarins, flavonoids, tannins, triterpenoids, anthraquinone, alkaloids, and saponins	Standard chemical tests and TLC	[15]
	Ethanol	Alkaloids, flavonoids, quinines, steroids, phenols, amide, alcohol, halogen compounds, and tannins	Standard chemical tests and FTIR	[20]
	Petroleum ether	Alkaloids, flavonoids, quinines, steroids, tannins, carbohydrates, vitamin C, and terpenoids	Standard chemical tests	[20]

**Table 3 plants-13-01096-t003:** Compounds identified from petroleum ether whole-plant extracts of *C. tridens* L. using GC-MS.

	Name of Compound	Molecular Formula	Molecular Weight	Chemical Structure	Detection Method
**1**	1-Iodo-2-methylundecane	C_12_H_25_I	296	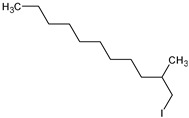	GC-MS
**2**	Heptadecane, 2,6,10,15-tetramethyl	C_21_H_44_	296	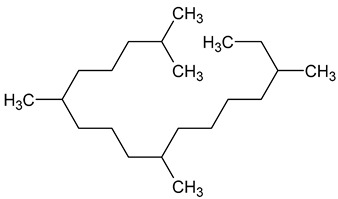
**3**	Heptadecane, 2,6-dimethyl	C_19_H_40_	268	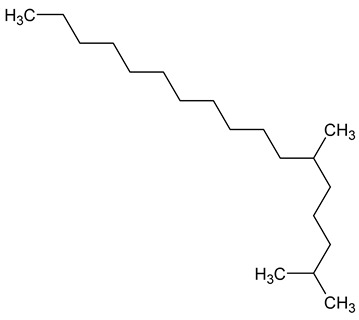
**4**	Sulphurous acid, hexyl tetradecyl ester	C_20_H_42_O_3_S	362	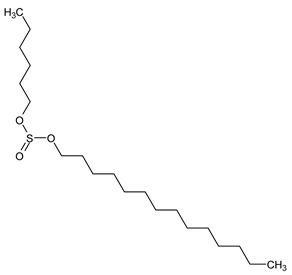
**5**	Hexadecanoic acid, 15-methyl, methyl ester	C_18_H_36_O_2_	284	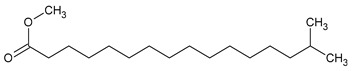
**6**	Eicosanoic acid, ethyl ester	C_22_H_42_O_3_	340	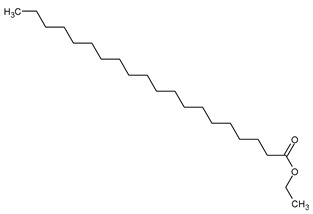
**7**	n-Hexadecanoic acid	C_16_H_32_O_2_	256	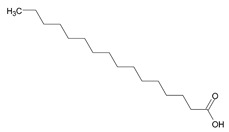
**8**	Heptacosane, 1-chloro	C_27_H_55_Cl	414	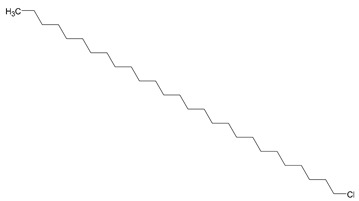
**9**	11,14-Eicosadienoic acid, methyl ester	C_21_H_38_O_2_	322	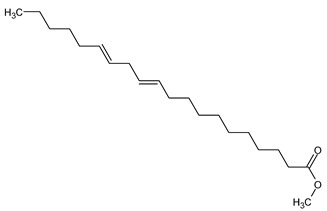
**10**	Phytol	C_20_H_40_O	296	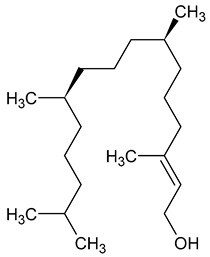
**11**	9-Octadecynoic acid	C_18_H_32_O_2_	280	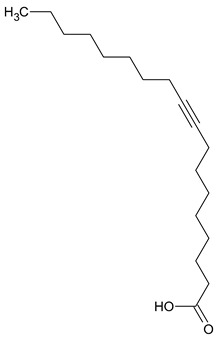
**12**	Tetracontane, 3,5,24-trimethyl	C_43_H_88_	604	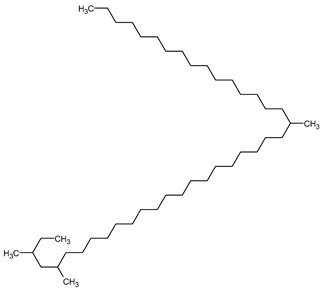
**13**	Tritetracontane	C_40_H_82_	604	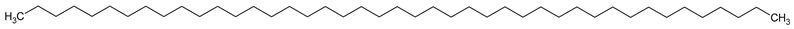
**14**	Sulphurous acid, hexyl pentadecyl ester	C_21_H_44_O_3_S	376	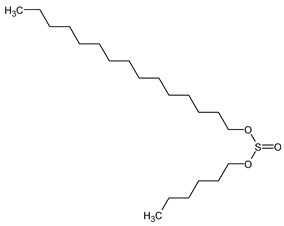
**15**	9,12,15-Octadecatrienoic acid, 2,3- bis(acetyloxy)propyl ester, (Z, Z, Z)	C_18_H_30_O_2_	436	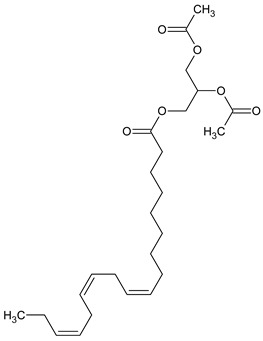
**16**	Squalene	C_30_H_50_	410	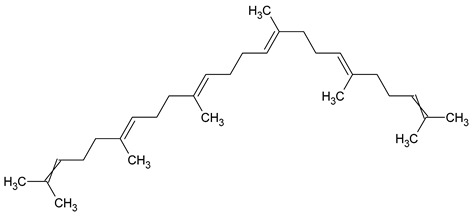
**17**	Sulphurous acid, butyl tridecyl ester	C_17_H_36_O_3_S	320	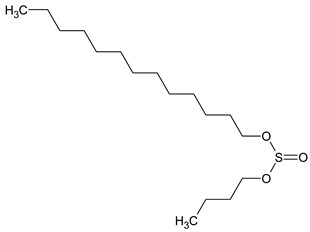
**18**	dl-alpha-Tocopherol	C_29_H_50_O_2_	430	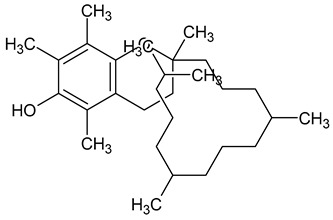
**19**	Stigmastan-6,22-dien, 3,5-dedihydro	C_29_H_46_	394	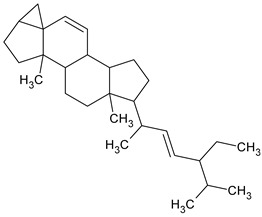
**20**	Cholesterol 3-O-[(2-acetoxy) ethyl]	C_31_H_52_O_3_	472	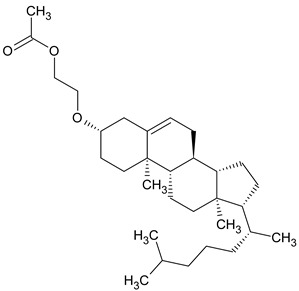
**21**	β-amyrin	C_30_H_50_O	426	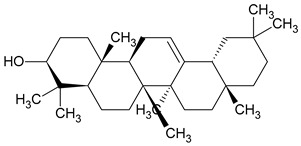

Complied using: ACD/ChemSketch (Freeware) 2023.1.2.

**Table 4 plants-13-01096-t004:** Nutritional content of *Corchorus tridens* L.

Plant Part	Nutritional Content	Amounts	Method of Analysis	References
Leaves	Total protein	3.74%	Micro-Kjeldahl nitrogen methods	[28]
Leaves		2.7–19.0%	Pico-Tag system	[23]
Leaves	Carbohydrates	7.29 ± 0.07%	Standard analytical methods	[24]
Leaves	Crude fibre	8.40 ± 0.51%	Standard analytical methods	[24]
	Fatty acids			
Leaves	16:0	4.65–4.70 mg/g	GC-MS	[23]
Leaves	18:0	0.46–0.47 mg/g	GC-MS	[23]
Leaves	18:1n9	0.46–0.64 mg/g	GC-MS	[23]
Leaves	18:2n6	2.13–3.10 mg/g	GC-MS	[23]
Leaves	18:3n3	7.79–10.7 mg/g	GC-MS	[23]
Leaves	Amino acids		Pico-Tag system	[23]
	Isoleucine	8.26–10.9 mg/g		
	Valine	12.0–15.3 mg/g		
	Phenylalanine	9.94–14.0 mg/g		
	Tyrosine	8.93–11.4 mg/g		
	Lysine	9.86–12.8 mg/g		
	Threonine	7.03–11.0 mg/g		
	Methionine	2.24–2.58 mg/g		
	Cysteine	4.08–4.87 mg/g		
	Tryptophan	8.21–11.3 mg/g		
	Aspartate	15.6–22.5 mg/g		
	Glutamate	23.1–29.7 mg/g		
	Serine	7.09–10.7 mg/g		
	Glycine	8.55–12.1 mg/g		
	Histidine	3.97–5.83 mg/g		
	Arginine	14.2–19.5 mg/g		
	Alanine	12.0–14.2 mg/g		
	Proline	17.4–19.7 mg/g		
	Leucine	15.5–21.3 mg/g		
Leaves	Ca	10.200–12.000 µg/g	ICP-AES	[23]
Leaves		14,759.0 ± 2139.0 ppm	NAA	[22]
Leaves		17,765.0 ± 374.0 ppm	EDXRF	[22]
Stems		12,350.0 ± 1803 ppm	NAA	[22]
Roots		3788.0 ± 591.0 ppm	NAA	[22]
Leaves	P	2.839–4.350 µg/g	ICP-AES	[23]
Leaves	K	3850 mg/kg	Flame photometry	[28]
Leaves		32.600–54.800 µg/g	ICP-AES	[23]
Leaves		43,120.0 ± 517.0 ppm	NAA	[22]
Leaves		21,590.0 ± 1884.0 ppm	EDXRF	[22]
Stems		54,290.0 ± 326 ppm	NAA	[22]
Stems		13,210.0 ± 349.5 ppm	EDXRF	[22]
Roots		21,110.0 ± 1457.0 ppm	NAA	[22]
Roots		8985.0 ± 158.5 ppm	EDXRF	[22]
Leaves	Cu	9.8–12.8 µg/g	ICP-AES	[23]
Leaves		114.885 ± 11.2 ppm	EDXRF	[22]
Stems		32.274 ± 2.14 ppm	EDXRF	[22]
Roots		22.457 ± 13.7 ppm	EDXRF	[22]
Leaves	Fe	69 mg/kg	Atomic absorption spectrophotometer	[28]
Leaves		385 µg/g dry weight	ICP-AES	[23]
Leaves		1130.0 ± 73.0 ppm	NAA	[22]
Leaves		8945.0 ± 89.34 ppm	EDXRF	[22]
Stems		8005.0 ± 167.0 ppm	EDXRF	[22]
Roots		376.0 ± 45.0 ppm	EDXRF	[22]
Roots		8410.0 ± 194.6 ppm	NAA	[22]
Leaves	Se	9.7–14.9 µg/g	ICP-AES	[23]
Leaves	Zn	<5.0 µg/g	ICP-AES	[23]
Leaves		51.0 ± 4.0 ppm	NAA	[22]
Stems		22.0 ± 3.0 ppm	NAA	[22]
Roots		14.0 ± 3.0 ppm	NAA	[22]
Leaves	Ba	138.0 ± 19.0 ppm	NAA	[22]
Leaves		8.825.0 ± 2.1 ppm	EDXRF	[22]
Stems		111.0 ± 18.0 ppm	NAA	[22]
Roots		BDL	NAA	[22]
Leaves	Mn	63.2 ± 2.6 ppm	NAA	[22]
Leaves		496.224 ± 39.3 ppm	EDXRF	[22]
Stems		32.0 ± 1.0 ppm	NAA	[22]
Stems		234.76 ± 15.7 ppm	EDXRF	[22]
Roots		23.0 ± 1.0 ppm	NAA	[22]
Roots		196.92 ± 25.2 ppm	EDXRF	[22]
Leaves		49.7–49.8 µg/g	ICP-AES	[23]
Leaves	Mg	3.85–4.78 µg/g	ICP-AES	[23]
Leaves	β-Carotene	47.00 mg/kg	Spectrophotometric method	[28]
Leaves	Rb	23.0 ± 2.0 ppm	NAA	[22]
Stems		20.0 ± 3.0 ppm	NAA	[22]
Roots		3.9 ± 0.8 ppm	NAA	[22]
Leaves	Co	0.47 ± 0.11 ppm	NAA	[22]
Stems		0.36 ± 0.12 ppm	NAA	[22]
Roots		BDL	NAA	[22]
Leaves	Ash content	4.27 ± 1.6 mg/100 g	Standard analytical methods	[24]
Leaves	Moisture content	67.62 ± 0.41%	Standard analytical methods	[24]
Leaves	Vitamin A	345.84 ± 0.21 mg/100 g	Spectrophotometric methods	[24]
Leaves	Vitamin B1,	0.68 ± 0.35 mg/100 g	Spectrophotometric methods	[24]
Leaves	Vitamin B2	1.072 ± 0.22 mg/100 g	Spectrophotometric methods	[24]
Leaves	Vitamin B3	0.72 ± 0.02 mg/100 g	Spectrophotometric methods	[24]
Leaves	Vitamin C	178. 83 ± 5.7 mg/100 g	Iodine methods	[24]

BDL, below determination level.

**Table 5 plants-13-01096-t005:** Pharmacological activities of crude extracts of *C. tridens* L.

Plant Part	Solvent Used	Pharmacological Activity	Bioassay Model	Results	Reference Standard	Reference
**Leaves**	Methanol	In vitro Anti-oxidant	DPPH	IC_50_ = 34.38 mg/mL	Vitamin C	[18]
	Ethyl ether	In vitro anti-bacterial	Disk diffusion (*S. aureus*)	0.67 I/C ^a^ and 0.92 I/P ^a^	Chloramphenicol and penicillin	[19]
			Disk diffusion (*E. coli*)	0.79 I/C ^a^ and 0.89 I/S ^a^	Chloramphenicol and streptomycin	
		In vitro anti-fungal	Disk diffusion (*C. albicans*)	0.92 I/M ^a^	Mycostatin	
	Alcohol	In vitro anti-bacterial	Disk diffusion (*S. aureus*)	0.71 I/C ^a^ and 0.82 I/P ^a^	Chloramphenicol and penicillin	
			Disk diffusion (*E. coli*)	0.89 I/C ^a^ and 0.94 I/S ^a^	Chloramphenicol and streptomycin	
		In vitro anti-fungal	Disk diffusion (*C. albicans*)	0.73 0.92 I/M ^a^	Mycostatin	
**Whole plant**	Ethanol	In vitro anti-bacterial	Disk diffusion (*S. aureus*)	14 mm	Chloramphenicol	[20]
			Disk diffusion (*B. subtilis*	14 mm		
			Disk diffusion (*K. aerogenes*)	14 mm		
			Disk diffusion (*E. coli*)	12 mm		
	Petroleum ether		Disk diffusion (*S. aureus*)	14 mm		
			Disk diffusion (*B. subtilis*)	26 mm		
			Disk diffusion (*K. aerogenes*)	13 mm		
			Disk diffusion (*E. coli*)	23 mm		
**Leaves**	Not indicated	In vitro Pesticidal properties	Has antibiosis effect on larval growth and development of *S. oliqua*	3.3 ± 5.8 to 154.0 ± 13.1 mg	Not indicated	[21]

a = ratio of diameters of the zones of inhibition to leaf extracts (10 µg) under observation (I) and the diameter of the zone of inhibition due to the reference standard’s anti-microbial substance. C = chloramphenicol (30 µg) against *S. aureus* 30 mm and against *E. coli* 32 mm. P = penicillin (10 units) against *S. aureus* 32 mm. S = streptomycin (10 µg) against *E. coli* 20 mm. M = mycostatin (100 units) against *C. albicans* 32 mm.

## Data Availability

Not applicable.

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
