# Peer review of "Corchorus tridens L.: A Review of Its Botany, Phytochemistry, Nutritional Content and Pharmacological Properties"

_plants, 2024, doi:10.3390/plants13081096_

Round 1

Reviewer 1 Report

Comments and Suggestions for Authors

This article is devoted to an overview of the C. tridens L plant and its biological potential. The main question that the authors consider is related to the availability and lack of information that could improve the use of this plant in the future, and further use it as a basis for creating a pharmacological agent. The authors have provided comprehensive information, the article is well constructed, clearly and clearly written. This review brings together all the available information, and the conclusions made in this review clearly identify knowledge gaps and the direction of further study for scientists working on this plant. Before accepting an article for publication, I would advise authors to make a few minor revisions:

1. It will be more convenient if the structures of compounds found in the plant will transfer to a table

2. If possible, add information about the percentage of each compound in the extract of C. tridens L

  Comments on the Quality of English Language

Minor editing of English language required

Author Response

  1. It will be more convenient if the structures of compounds found in the plant transfer to a table. The structures have been transferred to Table 3
  2.  If possible, add information about the percentage of each compound in the extract of C. tridens L. This comment is noted; however, the papers reviewed did not indicate the percentages of the identified compounds.  

Reviewer 2 Report

Comments and Suggestions for Authors

In this manuscript, the authors reviewed botany, phytochemistry, 2 nutritional content and pharmacological properties of Corchorus tridens. The contents of manuscript is comprehensive, and conforms to the scope of the journal; The review has certain reference significance for the development and utilization of C. tridens. However, there are logical and format problems in the writing of this manuscript, and shall be carefully revised. Moreover, “2.3 Phytochemical content” and “2.7Compounds identified in C. tridens L., extracts”; it is suggested that both expressions be combined for its readability.

Author Response

1. "2.3 Phytochemical content” and “2.7Compounds identified in C. tridens L., extracts”; it is suggested that both expressions be combined for its readability. This comment is noted and the phytochemical content and compounds identified have been combined. 

Reviewer 3 Report

Comments and Suggestions for Authors

Please correct the heading "results and discussion" and "Materials and Methods", more appropriate for an original article.

The column "leaves" of Table 3 is not coupled with "methods of analysis" and "references".

There are similar corrections to carry out in tables 4-5, as well.

Regarding Figures 2 and 3, the structures were not prepared in an uniform way. I suggest to download SMILES from PUBCHEM and generate the figure from SMILES through an appropriate program (for instance ChemSketch, ChemDraw, ...). This also will permit to avoid errors, as in the case of the structure #15 in figure 3.

Please make uniform the references in the reference section.

Author Response

  1. Please correct the heading "results and discussion" and "Materials and Methods", more appropriate for an original article. -This has been revised.
  2. The column "leaves" of Table 3 is not coupled with "methods of analysis" and "references"- This was due to the lack of spacing, and it has been inserted.
  3. There are similar corrections to carry out in tables 4–5 as well. These have also been corrected, please see table 4 to 5.
  4. Regarding Figures 2 and 3, the structures were not prepared in an uniform way. I suggest to download SMILES from PUBCHEM and generate the figure from SMILES through an appropriate program (for instance ChemSketch, ChemDraw, ...). This also will permit to avoid errors, as in the case of the structure #15 in figure 3.- This has been addressed and all the structures were reconstructed using ACD/ChemSketch software using Isomeric SMILES notation from PUBCHEM.
  5. Please make uniform the references in the reference section. This has been address please refer to the reference section to see the changes effected.

Reviewer 4 Report

Comments and Suggestions for Authors

Dear authors:

I have carefully revised your paper and I think that it is good enough to be published in this journal. However, I think that few littles’ things should be considered before. I have included my suggestions throughout the paper.

Best wishes.

Jesús.

Author Response

  1. Where are 6 and 7? Please revise the reference order- The order has been revised.
  2. Reference formatting according to journal specifications- This has been revised throughout the document.
  3. Please provide the author of each species- The references has been provided for each species see line 58-62.
  4. olitorius, C. capsularis, C. trilocularis, and C. depressus- The genus name has been added in these species see line 63-64.
  5. et al on italic- et al has been italicised throughout the entire document.
  6. I think that you could include the complete list of synonyms- A complete list of synonyms has been included see line 106-108.
  7. I think that you should mention that it is widely distributed in Africa although it also appears in India an Australia- This has been mentioned see line 149-150.
  8. quinOnes???- quinines changed to quinOnes see line 197.
  9. Although (double spacing)- Double spacing removed see line 350.
  10. Change tertbutylhydroquinine to tertbutylhydroquinOne- This has been changed, see line 352.
  11. Remove the irregular spacing- This has been removed, see line 370-374.
  12. Replace n-hexadecanoic with n-Hexadecanoic- This has been done.
  13. I agree with this section although I think that it could be interesting if you separated it paragraphs. The paragraphs have been separated and there is one for the conclusion and another one for the future perspectives.
  14. Consistency in the list of references section: This has been attended to and those written in upper case have been changed to sentence case.